# Vector-borne diseases and the Syrian conflict: A systematic review of literature from Syria and neighbouring, refugee-hosting countries

Jack Carew[1], Remi Simpson[2], Timesh D. Pillay[2], Angel Desai[3], Richard Toalster[4], Pylin Parkes[2], Aula Abbara[2,5]*

**1** Yale University School of Public Health, Yale University, New Haven, Connecticut, United States of America, **2** Imperial College School of Medicine, Imperial College London, London, United Kingdom, **3** Department of Internal Medicine, University of California, Davis, California, United States of America, **4** Independent Consultant, London, United Kingdom, **5** Syria Public Health Network, London, United Kingdom

* a.abbara15@ic.ac.uk

## Abstract

### Objectives

Syria's conflict has forced more than half its populations from their homes, decimated its health system and water and sanitation infrastructure, leading to an increase in communicable diseases, particularly vector-borne diseases (VBDs.) This systematic review explores the epidemiological burden and geographical distribution of VBDs in Syria and neighbouring, refugee-hosting countries following the onset of the Syrian uprisings.

### Methods

Four databases were searched using appropriate MeSH terms. Included studies reported on data collected between March 2011 and May 2024 on VBD-affected populations in Syria, Turkey, Jordan, Lebanon or Iraq. Data were extracted, study quality assessed, and findings synthesized in narrative form.

### Results

33 studies were included; all but one reported on leishmaniasis, the exception reported on malaria. 16 were from Turkey, nine from Lebanon, five from Syria and three from Jordan. Data showed an increase over time in the numbers of reported leishmaniasis cases and geographical spread as well as barriers to healthcare access for Syrian refugees.

### Conclusions

In this systematic review, Leishmania was the primary VBD described in this context. Studies from country-specific health ministries mostly reported data whereas smaller

**Data availability statement:** All data is available in the manuscript.

**Funding:** The author(s) received no specific funding for this work.

**Competing interests:** The authors have declared that no competing interests exist.

studies added more granular information including around healthcare access. Additional studies are needed to identify vector reservoir populations and to investigate the burden of other VBDs in this region.

## Author summary

The conflict in Syria has severely impacted public health infrastructure, leading to a rise in communicable diseases, particularly vector-borne diseases (VBDs) such as leishmaniasis. This systematic review examined 33 studies published between March 2011 and May 2024, focusing on Syria and neighboring refugee-hosting countries—Turkey, Lebanon, Jordan, and Iraq. Leishmaniasis emerged as the most commonly reported VBD among Syrians, with only one study addressing malaria. Reported cases of leishmaniasis increased over time and spread geographically, especially in the surrounding refugee-hosting areas. However, barriers to healthcare access and poor disease surveillance suggest the true burden is likely underestimated. Most studies relied on health ministry data, while smaller studies highlighted critical issues such as healthcare inaccessibility. There is a notable gap in research which takes a One Health approach and on effective intervention strategies. Further research on emerging or re-emerging VBDs is essential given changes in the post-conflict period including population movements, reduced funding for vector control and changes in climate in the region.

## Background

The Syrian conflict began with uprisings in March 2011 and escalated to violent conflict by mid-2012 [1]; it has displaced more than half the country's pre-war population of 22 million [2]. There are over 6.4 million Syrian refugees as a result, with a further 7.2 million who have been internally displaced (IDPs) [2,3]. The majority of Syrian refugees have settled in neighbouring countries, including Turkey, Jordan, Lebanon and Iraq [4,5]. (See Fig 1) The conflict has caused widespread destruction of infrastructure including healthcare, water and sanitation and hygiene (WASH), buildings and electricity [6,7–10]; 70% of the Syrian population do not have access to adequate drinking water [1] and more than 80% live in poverty [6,11]. In December 2024, the Syrian regime was toppled which has ushered in a new era in Syria with changes to population movements (including returnees from surrounding refugee hosting countries), conflict dynamics and impacts on healthcare access, vector control and surveillance.

There is an established literature on humanitarian crises and vector-borne diseases (VBDs), with forcible displacement of communities exacerbating transmission or even introducing communicable diseases to new regions [12]. Prior conflicts have shown that population movement and breakdown of prevention measures can lead to

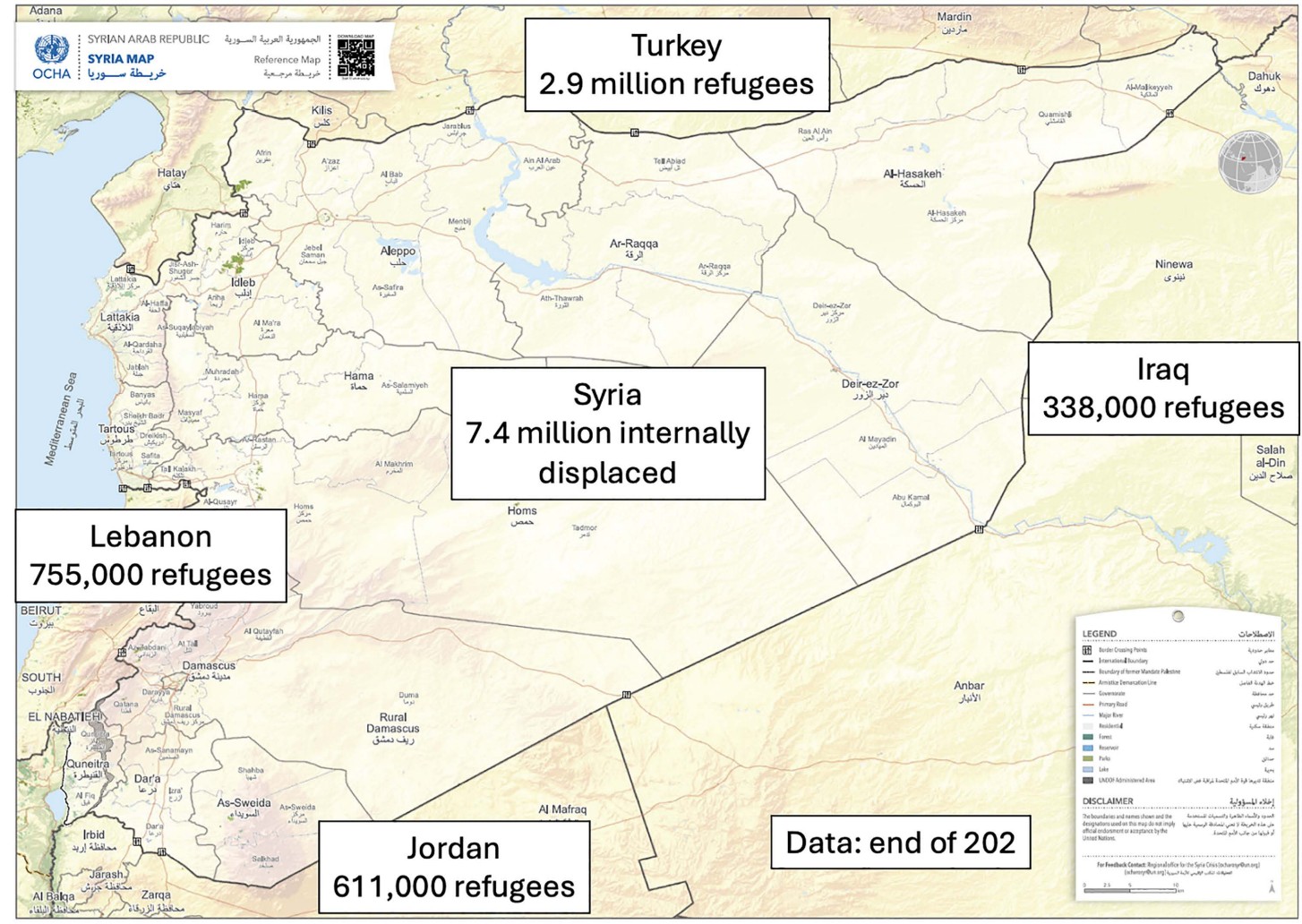

**Fig 1. A map showing the scale of the Syrian refugee crisis, with mass internal displacement and refugee-migration.** UNHCR figures as of May 2020. IDMC figures as of December 2019. IDPs = Internally Displaced Persons. The administrative boundaries and surface map of Fig 1 come from Open Data Soft (https://public.opendatasoft.com/explore/assets/world-administrative-boundaries/), and the license allows for academic use (https://www.nationalarchives.gov.uk/doc/open-governmentlicence/version/3/).

epidemics, for example, malaria and dengue in Yemen [13], and leishmaniasis in Syria, Sudan, and Iraq [14]. VBDs have been endemic in the region for many years [15,16]. The cutaneous form of leishmaniasis, a parasitic VBD transmitted by the *Phlebotomus* sandfly, is endemic to Syria [14]. Before the conflict, vector control interventions with insecticide-treated nets in Syria had successfully reduced incidence of cutaneous leishmaniasis (CL) [17,18] and reports indicated that previous vector spraying for malaria had also caused a decrease in CL incidence [19]. Pre-conflict incidence of CL in Syria was reported as 23,000 cases/year in 2010 though this is likely to be 3–5 times higher due to under-reporting; the epidemiological burden increases in the post-conflict period with more than 53,000 cases reported as early as 2012 [20]. Reports of visceral leishmaniasis (VL) are rare but appear to be increasing [21–23]. Inside Syria, a number of humanitarian organizations support Leishmania investigation and treatment including the Mentor Initiative which commenced its program in September 2013 [24,25].

Despite the impact of conflict on VBDs and their relative importance in this region, there has been little comprehensive analysis of literature of VBDs in Syria and neighbouring refugee-hosting countries after the onset of conflict [26,27]. This systematic review aims to explore clinical-epidemiological literature relevant to VBDs within Syria and in the main neighbouring refugee-hosting countries, and to identify factors related to VBDs and the healthcare system relevant to displaced Syrians or Syrian refugees. Given that this is a systematic review, we have not made a hypothesis as our aim is to capture all relevant literature and undertake a narrative review.

## Methods

A systematic review of academic literature describing the burden of VBDs and clinical data in Syria and neighbouring refugee-hosting countries (Iraq, Lebanon, Jordan, Turkey) between March 2011 and 20th May 2024 was performed. The Preferred Reporting Items for Systematic Reviews and Meta-analyses Protocols (PRISMA-P) was used to develop this review protocol. See Fig 2.

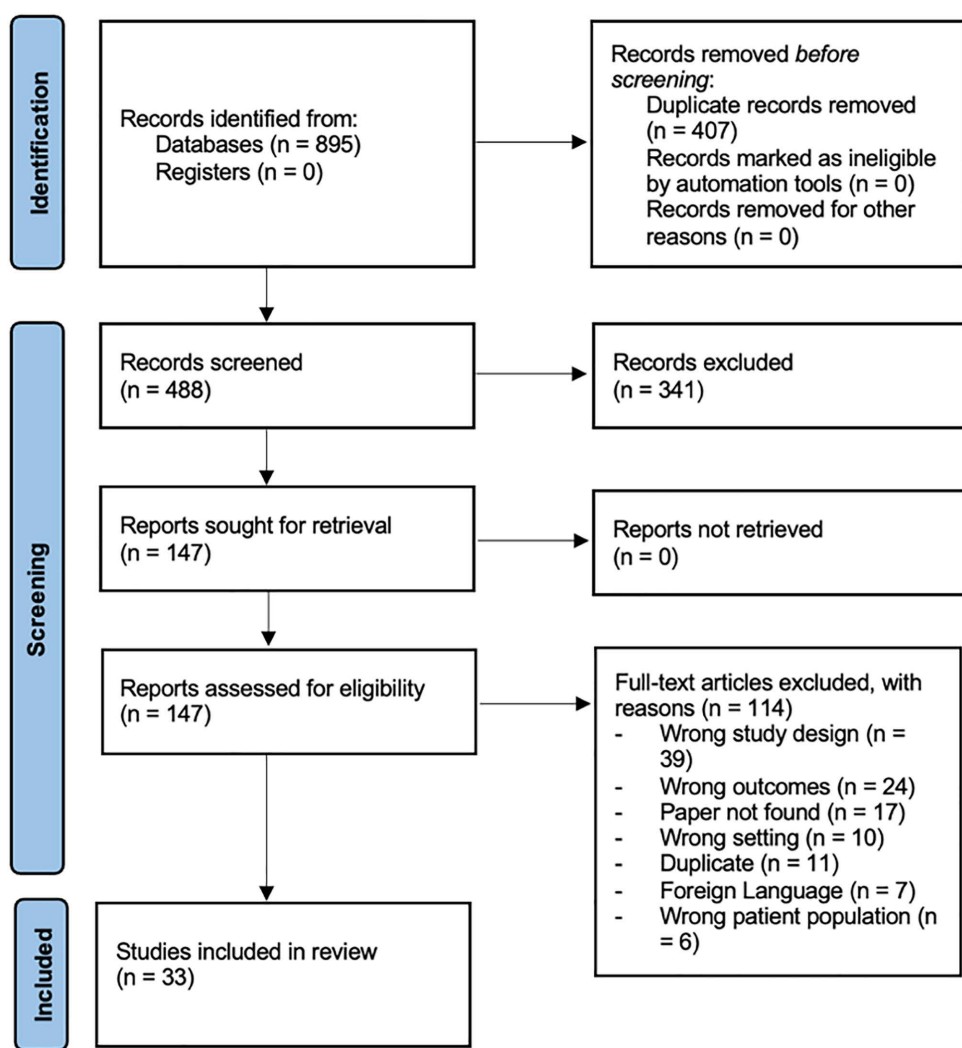

**Fig 2. Prisma Flow Diagram showing the number of included studies and excluded records [28].**

## Search strategy

Four databases were used to perform a literature search. Ovid was used to search Medline, Embase and Global Health platforms. Scopus was searched independently. Search terms were constructed using country names (Iraq, Jordan, Lebanon and Turkey) paired with the terms "refugee" and "migrant" alongside collective and specific terms for VBDs as defined by the World Health Organization published list (see Table 1). Medical subject headings (MeSH terms) were used, and terms and concepts were combined with Boolean operators. (See S1 Appendix for full search terms).

All diseases in the WHO VBDs list were included for a comprehensive search, allowing identification of any unexpected diseases that are not endemic to the region. The term 'conflict' and related terms were not used in the initial search strategy to ensure that all data were captured even if the reports did not specifically mention conflict. Grey literature was excluded from the search to standardize data quality and data extraction process. We defined the burden of disease in terms of case numbers and epidemiology.

**Table 1. WHO list of vector-borne diseases, according to their vector [29].**

| Vector | Disease caused | Type of pathogen |
|---|---|---|
| **Mosquito** | | |
| Aedes | Chikungunya | Virus |
| Aedes | Dengue | Virus |
| Aedes | Lymphatic filariasis | Parasite |
| Aedes | Rift Valley fever | Virus |
| Aedes | Yellow Fever | Virus |
| Aedes | Zika | Virus |
| Anopheles | Lymphatic filariasis | Parasite |
| Anopheles | Malaria | Parasite |
| Culex | Japanese encephalitis | Virus |
| Culex | Lymphatic filariasis | Parasite |
| Culex | West Nile fever | Virus |
| Aquatic snails | Schistosomiasis (bilharziasis) | Parasite |
| Blackflies | Onchocerciasis (river blindness) | Parasite |
| Fleas | Plague (transmitted from rats to humans) | Bacteria |
| Fleas | Tungiasis | Ectoparasite |
| Lice | Typhus | Bacteria |
| Lice | Louse-borne relapsing fever | Bacteria |
| Sandflies | Leishmaniasis | Parasite |
| Sandflies | Sandfly fever (phlebotomus fever) | Virus |
| Ticks | Crimean-Congo haemorrhagic fever | Virus |
| Ticks | Lyme disease | Bacteria |
| Ticks | Relapsing fever (borreliosis) | Bacteria |
| Ticks | Rickettsial diseases (e.g., spotted fever and Q fever) | Bacteria |
| Ticks | Tick-borne encephalitis | Virus |
| Ticks | Tularemia | Bacteria |
| Triatome bugs | Chagas disease (American trypanosomiasis) | Parasite |
| Tsetse flies | Sleeping sickness (African trypanosomiasis) | Parasite |

## Study selection

Search results were imported into Covidence, a web-based data-collection platform. Duplicates were automatically removed. Three reviewers (RS, PP, JC) then applied the inclusion and exclusion criteria to the titles and abstracts of papers identified in the database searches.

The time period selected was between 15th March 2011, the date the Syrian uprisings began, and 20th May 2024, the date of the final database searches. Letters to the editor, reviews of the literature and single case reports were excluded due to their minimal potential for contribution to a wide overview of VBDs burden. Full eligibility criteria are listed in Table 2. Eligible papers then underwent full text review. References of the selected papers were checked to identify relevant studies not captured by the primary search.

## Data analysis

Basic study data (e.g., dates of study, geographical location) were extracted into a Microsoft Excel spreadsheet. These are summarized in Table 3. Relevant themes were then identified in an iterative process before official extraction of data from each study (e.g., diseases reported, origin of study population). Qualitative data were also extracted to facilitate a narrative synthesis of the emerging themes.

## Assessment of quality of evidence

Quality assessment was performed using 5 components from the Effective public health practice project (EPHPP) quality assessment tool for quantitative studies [31]. These items allowed assessment of study method (component A: selection bias, component B: study design, component E: data collection methods) and reporting of study results (component H: analyses), as well as a global rating. Numerical ranking for all components (1–3) correlated with strong, moderate, or weak ratings; sum scores were used to form a global rating of the paper. See S2 Appendix.

## Results

After assessment of 859 abstracts and removal of duplicates, 145 full-text papers were screened. Finally, 33 studies were selected for inclusion and underwent analysis. Table 2 details the inclusion process.

**Table 2. Eligibility criteria.**

| Criteria | Included | Excluded |
|---|---|---|
| Population | Syria<br>Neighboring Syrian refugee-hosting countries (Lebanon, Jordan, Turkey, Iraq) [5] | Syrians residing in any other country |
| Intervention | Surveillance of vector-borne diseases included in the WHO vector-borne diseases list [30] (refer to Table 1) | Surveillance of diseases not included in the WHO vector-borne diseases list |
| Control | Any | No limit |
| Outcomes | Epidemiological descriptions of disease, case numbers, vector speciation (for diagnostic and epidemiologic purposes) | Vaccination data, data with no clinical interest |
| Study Design | Retrospective and perspective studies, case series, cohort studies, case-control studies, randomised controlled trials, non-randomised trials | Single case reports, reviews of the literature, letters to the editor |
| Time Period | March 2011 – 20th May 2024 | Outside of this period |
| Language | All where full text is available in English | Papers where full text is not available in English |

**Table 3. Included studies.**

| Manu-script number | First author, publi-cation year, paper reference(s) | Study type | Study Period | Reporting Country | Region (where specified) | Populations included | Number of patients |
|---|---|---|---|---|---|---|---|
| 1 | Kocarslan, 2013, 34 | Retrospective obser-vational study | Nov 2012- March 2014 | Turkey | Sanliurfa city, south-eastern | Turkish (30) and Syrian (24) | 54 |
| 2 | Saroufim, 2014, 35 | Retrospective obser-vational study | Nov 2012- Feb 2013 | Lebanon | Bekaa (35.4%), Tripoli (26.2%), Akkar (18.2%), Beirut (18.2%). | Syrian (67% from Aleppo, 27% from Homs, 5% from Damascus) | 1275 |
| 3 | Alawieh, 2014, 15 | Retrospective Observational Study | January 2001 - March 2014 | Lebanon | Bekaa (38/100,000); mount Lebanon (19/100,000); Northern areas (17/100,000); Southern areas (6/100,000) | Syrian (96.6%). Leba-nese nationals/ Palestin-ian refugees (3.4%). | 1033 |
| 4 | Koltas, 2014, 36 | Cross sectional study | July 2003 and July 2013 | Turkey | Southern Turkey | Syrian & Turkish | 280 |
| 5 | Turan, 2015, 37 | Cross sectional study; comparative | January 2012 - January 2013 | Turkey | Sanliurfa city, south-eastern | 685 Syrian & 685 Turk-ish patients | 1370 |
| 6 | Inci, 2015, 38 | Retrospective Observational Study | January 2011 - June 2014 | Turkey | – | Turkish & Syrian (76, 69%) | 110 |
| 7 | Zgheib, 2016, 39 | Cross sectional study | 2011-2015 | Lebanon | – | Syrian (165) & Leba-nese (4) | 169 |
| 8 | Dunya, 2016, 40 | Cross sectional study | Not stated | Lebanon | – | Syrian (163) & Leba-nese (5) | 168 |
| 9 | Alsaied, 2017, 41 | Retrospective obser-vational study | Jan 2014 - Dec 2015 | Syria | Northern Syria | Syrian | 46039 |
| 10 | Ozkeklikci, 2017, 42 | Cross sectional study | January 2009 - July 2015 | Turkey | Gaziantep, South-eastern Turkey | Turkish (88, 33%) & Syrian (174, 66%), Afghanistan (1) | 567 |
| 11 | Eksi, 2017, 43 | Cross sectional study | April 2014 - April 2015 | Turkey | South-eastern Turkey | Syrian (433) and Turkish (25) | 468 |
| 12 | Hawat, 2017, 44 | Retrospective obser-vational study | 2006-2016 | Syria | Lattakia governorate | Syrian | 172 |
| 13 | Kaman, 2017, 22 | Retrospective obser-vational study | January 2014 - December 2015 | Turkey | Ankara, central Turkey | 2 Turkish citizens, 14 Syrian refugees | 16 |
| 14 | Beyhan, 2017, 45 | Case series | 2014 | Turkey | Central Anatolia. | 2 Syrian, 1 Turkish patient | 3 |
| 15 | Rehman, 2018, 27 | Ecological study | September 2013–2018 | Syria | Northern Syria | Syrian | 64498 |
| 16 | Gurses, 2018, 46 | Cross sectional study | 2012-2014 | Turkey | Sanliurfa city, south-eastern | Not Specified | 135 |
| 17 | Hajj, 2018, 47 | Case series | 2014-2017 | Lebanon | Beirut | Northern Syria | 5 |
| 18 | Muhjazi, 2019, 28 | Retrospective obser-vational study | 2011-2018 | Syria | All of Syria | Syrian | 82275 |
| 19 | Youssef, 2019, 48 | Retrospective Observational Study | 2008–2016 | Syria | Lattakia | Syrian | 8168 |
| 20 | Amr, 2019, 49 | Retrospective cohort analysis | 2010-2016 | Jordan | – | Syrian & Jordanian | 1243 |
| 21 | Eroglu, 2019, 50 | Cross sectional study | 2013-2015 | Turkey | Gaziantep, South-eastern Turkey | Syrian 93.8% (845/900), Turkish 6.2% (55/900. | 900 |
| 22 | Özbilgin, 2019, 51 | Cross sectional study | January 2013 - December 2016 | Turkey | 18 provinces across Turkey, mainly southern | Not specified | 356 |

*(Continued)*

**Table 3.** (Continued)

| Manu-script number | First author, publi-cation year, paper reference(s) | Study type | Study Period | Reporting Country | Region (where specified) | Populations included | Number of patients |
|---|---|---|---|---|---|---|---|
| 23 | Karaosmanoglu, 2019, 52 | Retrospective obser-vational study | 2017-2018 | Turkey | – | Syrian | 117 |
| 24 | Hijawi, 2019, 53 | Cross sectional study | Not stated | Jordan | – | Jordanian (39) and Syrian (27) | 66 |
| 25 | Safadi, 2019, 54 | Cohort analysis | January - June 2017 | Lebanon | North Lebanon | Syrian | 48 |
| 26 | Özbilgin, 2019, 55 | Retrospective cohort study | 2012-2016 | Turkey | – | Syrian (30), Iraq (5), Afghanistan, Iran, Turkmenistan | 38 |
| 27 | Karakus, 2019, 56 | Cross sectional study | 2014-2018 | Turkey | South-eastern Turkey | Syrian | 25 |
| 28 | Alhawarat, 2020, 57 | Retrospective obser-vational study | 2010-2016 | Jordan | – | Syrian (559) and Jorda-nian (646) | 1243 |
| 29 | Yentur Doni, 2020, 58 | Cross-sectional study | November 2015 – Novem-ber 2017 | Turkey | Sanliurfa, Southeastern Turkey | Syrian (60), Turkish (94) | 154 |
| 30 | Bizri, 2021, 59 | Ecological study | 2005-2018 | Lebanon | – | Not specified | 4234 |
| 31 | Hammoud, 2022, 60 | Ecological Study | 2010-2019 | Lebanon | – | Not specified | 1976 |
| 32 | Farah, 2023, 61 | Ecological Study | 2013-2019 | Lebanon | – | Syrians, Lebanese | 6581 |
| 33 | Özbilgin 2023, 62 | Retrospective obser-vational study | 1996-2022 | Turkey | – | Not specified, refugees and asylum seekers (22) | 131 |

## Characteristics of included studies

Studies identified were published between 2013–2023. Of these, 12 were cross-sectional studies [32–43], 12 retro-spective observational studies [14,25,44–53], four ecological studies [24,54–56], three cohort analyses [57–59], and two case series [60,61]. 16 reported on populations in Turkey [32,33,36–40,42–45,47,51,53,59,60], nine in Lebanon [14,34,35,46,54–56,58,61], five in Syria [24,25,48–50], and three in Jordan [41,52,57]. None described populations in Iraq. 32 included papers focused on Leishmaniasis while one focused on malaria. See Fig 3.

Further details of these studies are provided in S3 Appendix.

Three major themes emerged from review of the included studies: 1. VBD related factors 2. Causes for the increased burden of VBDs and 3. Barriers to accessing healthcare for VBD affected displaced populations.

1. VBD related factors

### Disease types

All but one of the 33 eligible studies reporting on VBDs in Syria and neighbouring refugee-hosting countries focused solely on Leishmaniasis cases. 31 predominantly discussed CL burden [24,25,32–52,54–61]; five of these also reported on VL cases [14,24,36,38,45]. One case series of 3 patients in Central Anatolia, Turkey of two Syrian and one Turkish patient reported on VL [60]. One study reported on cases of imported malaria in Turkey [53]. None of the selected studies dis-cussed epidemiological or clinical findings for any other VBDs.

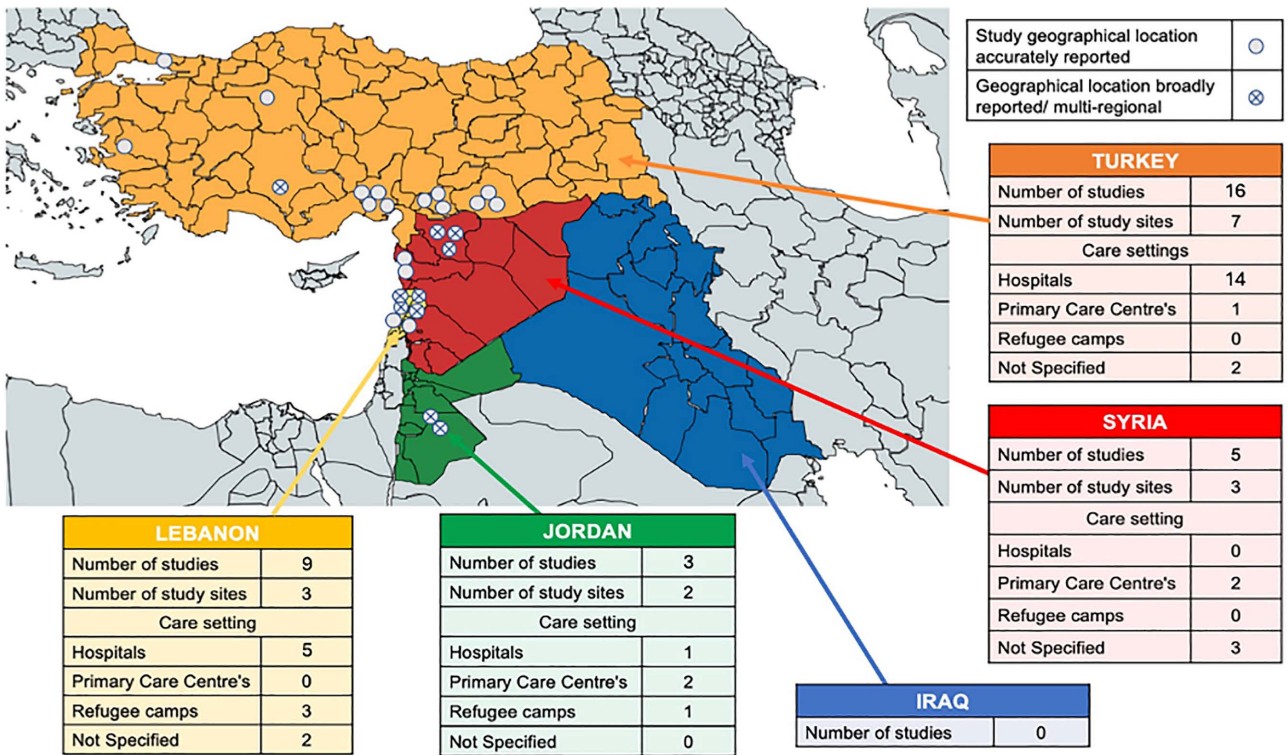

**Fig 3. The geographical location of included studies.** The administrative boundaries of Fig 3 were obtained from GADM (https://gadm.org/data.html), and the license indicates that the data are freely available for academic use (https://gadm.org/license.html).

## Speciation

Fifteen studies reported the CL causative species, confirmed by PCR [24,32,34–38,40–43,46,58–60]. One paper by Rehman *et al,* reported on molecular-epidemiologic survey data from the Mentor Initiative in northern Syria [24]. Of 99 specimens successfully shipped to Vienna, Austria, they were able to sequence 93. They detected *L.tropica* in 78% of samples and *L.major* in 13% of samples; *L.infantum* and *L.donovani* made up 9% of cases [24] Similar findings were reported from four studies based in Lebanon of between 48 and 1275 cases which found *L.tropica* to be the most prevalent, accounting for between 83% and 95% across the studies; the remainder (between 5–17%) were caused by *L.major* [34,35,46,58].

Nine studies based in Turkey, mostly in south-eastern areas where most Syrian refugees reside, also reported *L.tropica* to be the main causative species based on studies of 3–567 patients [36–38,40,42,43,59,60]. In seven of these, it accounted for between 84–100% of cases [36–38,40,42,43,60]. Of these studies, four papers identified *L.major* as the second most commonly identified species (2–21%) [41,43,49,57], while two found *L.infantum* to be the main causative species after *L.major* (10-36.5%) [35,46]. Two papers, notably both from Jordan, identified cases of CL caused by *L.donovani* [41,57] and one case of CL caused by *L.aethiopica* [41]. In contrast to the other countries, molecular analysis from Jordan reported a lower proportion of *L.major* at 69% with *L.tropica* accounting for 31% of cases [41]. Ozbilgin *et al* performed molecular investigations of 38 samples from Turkey and found evidence of a new *L.tropica* species population structure; this was a mixture of Syrian and Turkish species genomes [58].

Three studies reported on the causative species of reported cases of VL; of these, one study from Lebanon and one from Syria identified *L.infantum* as the cause in all patients tested [24,60]. A study from Turkey found 60% were infected by *L.infantum*, with *L.donovani* (30%) and *L.major* (10%) also represented [45].

One study in Turkey reported on the causative species of imported cases of malaria; of these 131 were *Plasmodium falciparum* [53].

**Disease burden**

Papers which presented MoH data for their analyses, reported on a larger number of cases and reported multi-year data giving a useful overview of trends. Muhjazi *et al* used data from three reporting systems to capture available data on leishmania within Syria between 2011 and 2018; their data set included 82275 cases from across Syria. They note that CL case numbers increased from 58,156 in 2011 to a peak of 86,269 in 2015; cases then dipped before rising to 82,275 in 2018 [25]. They also noted a four-fold rise in the number of leishmaniasis cases between 2013 and 2018 in Raqqa governorate in northeast Syria [25]. A retrospective analysis by Youssef *et al* focused on 8,168 in Latakia and included data from 2008 and 2016. They note a peak of CL in 2013, with 30% of the cases reported in that year and a subsequent decline from 2014 onwards [50]. Importantly, this study notes 24 cases of VL in Latakia between 2013 and 2014 [36,50]. Alsaied *et al* analyse retrospective data of 46,039 consultations from primary care centres run by a humanitarian organisation in northern Syria [48]. CL consultations accounted for most of the infectious diseases seen with a mean of around 1170 cases/ month/ centre [48]. Rehman *et al* also report data from northern Syria across five cities noting a peak of 7,599 CL cases/month in February 2015, falling to 2,476 cases/month in February 2016 after control programmes [24].

Other studies reported on single centre, regional or comparatively smaller datasets from Turkey, Lebanon and Jordan. Ozkeklikci's retrospective review of 567 cases in Gaziantep in southeastern Turkey between 2009 and 2015 noted a peak of 76 cases among Syrian patients in 2013 and only three by 2015 [36]. A study in Lebanon by Alawieh *et al* using Ministry of Public Health data analysed data on 1033 confirmed CL cases between 2008 and 2013 [14]. Almost all cases (96.6%) were among Syrian refugees and the highest proportion were in regions in Lebanon where the most refugees resided; in particular, the Beka'a valley had 38 per 100,000 cases compared to 6 per 100,000 cases in south Lebanon [14]. Of interest is the study by Amr *et al*, one of only three manuscripts from Jordan in our review. Of 558 Syrian patients with CL included between 2010 and 2016, 92% were classified as imported. Diagnoses of CL increased from 2012 onwards and the proportion of Syrian patients also increased from 4.4% in 2011 to 55.2% in 2016 [57]. Zarqa governorate where there are two camps which host Syrian refugees accounted for 21% of all cases [57].

2. Causes for the increased burden of VBDs

In countries neighbouring Syria which host large numbers of refugees, Syrian refugees accounted for most of the CL cases with increased prevalence coinciding with an increase in the numbers of refugees arriving in these countries. Nine studies based in Turkey reported on leishmaniasis in a mixed population of Turkish citizens and Syrian refugees settled in Turkey [32,33,36,39,44,45,47,48,60]; one study from Turkey investigated VBDs among refugees of whom 79% were Syrian [59] and three reported VBDs in Syrian refugees only [42,43,51]. Among Lebanese studies, six reported on data from Lebanese citizens and Syrian refugees [14,34,35,54–56]; all studies reporting from Jordan included refugee and local population cases [41,52,57] six studies from outside Syria (three in Lebanon, three in Turkey) reported data from Syrian refugees only [42,43,46,51,58,61].

Four studies from Lebanon which included Syrian refugees noted that the majority of patients had been forcibly displaced from Aleppo [28,34,35,54]; one study found 67.3% of 1275 patients from multiple refugee camps within the Bek'aa Valley had been forcibly displaced from Aleppo, with 27.3% from Homs [28]. However, a smaller study of 48 CL patients in north Lebanon found 44% of patients had been forcibly displaced from Idlib and 25% from Hama [58].

## Living conditions

Eight studies referenced deteriorating refugee camp conditions or substandard urban living conditions among these populations. Overcrowding, poor sanitation and limited access to treatment were identified as contributors to disease burden [14,44,46,48,50,52,54,55].

## Interventions to reduce exposure

Three studies referenced initiatives by the Lebanese Ministry of Public Health between 2013 and 2014 [14,24,55]. One identified that surveillance, treatment, and preventative measures are needed to limit outbreaks [14] and another that accelerated prevalence during the beginning of the armed conflict declined after implementation of a comprehensive control program by an international not-for-profit organisation was commissioned to implement an integrated leishmaniasis control program initiative [24].

However, additional papers discussed the need for better control initiatives and preventative measures to reduce CL cases [14,38,39]. Three papers also emphasised the need for increased *Leishmania* vector speciation and related studies [38,40,46].

3.  Barriers to accessing healthcare for VBD-affected displaced populations

Different barriers relating to healthcare access and health seeking behaviour for VBD investigation and treatment were noted. Three studies noted that CL underreporting was likely due to patients not seeking healthcare even if affected by a VBD. One noted that CL lesions could be small, painless, heal spontaneously or have an insidious onset; these factors could discourage health-seeking behaviour entirely or until the disease had progressed due to general challenges that refugees had in accessing healthcare [25,38,60]. It was also noted that patients could be discouraged from seeking treatment due to a scarcity of healthcare and treatment, or for safety reasons [25,38]. Stigma related to CL (which is associated with poverty and rural residence) was cited by one study as a reason for patients not seeking treatment [48]. The psycho-social impact of a disfiguring disease, such as CL, was discussed in one study from Lebanon, with possible implications on future mental health [58]. A study from Jordan emphasised the need for increased patient education and encouragement of early health-seeking behaviour to improve disease management [57].

Sixteen studies documented clinical data from hospital settings [32,33,36,37,42–45,47,51,53,57–61]; 13 out of 16 studies based in Turkey were from hospital settings [32,33,36,37,42–45,47,51,53,59,60]. All studies in Jordan and two reporting from Lebanon included data collection from refugee camps [34,35,41,46,52]. See Fig 3. Other noted barriers are included in S3 Appendix.

## Quality of included studies

Five of the included studies were given a methodological quality grading of strong, 24 moderate and four poor. Common reasons for a grading of poor were possible selection bias, no controlling for confounders or poor study design. See S3 Appendix for a detailed summary.

## Discussion

This systematic review of VBDs in Syria and neigbouring, refugee-hosting countries identified academic literature related only to leishmaniasis with most reporting on CL rather than VL. The burden and geographical distribution of cases, increases in annual trends of CL and factors relating to healthcare access for Syrian refugees and IDPs were noted. There have been reports of other VBDs affecting humans in the included countries [62] however their absence in the academic literature after the onset of the Syrian uprisings likely represent the predominance of leishmaniasis both in visibility (particularly for CL) and disease burden (28,000/year at the onset of conflict) which far outnumbers other VBDs affecting humans in the countries of the study [24,63].

We note for example, an absence of literature in Syria or among Syrian refugees in neighbouring countries on Crimean Congo Haemorrhagic Fever (CCHF,) and other tick-borne infections, e.g., rickettsial infections or arboviruses. For CCHF, this is important given its prevalence in Turkey and Iraq, both of which share borders with Syria. This review forms a baseline to understand the current availability of data on VBDs (or lack thereof) and, given the impacts of climate change how VBDs may emerge or re-emerge; this would also include the impact on leishmaniasis, whose distribution in Syria has changed over the course of the conflict and is itself, impacted by climate change [64,65].

## Increased burden and changes to the geographical distribution of leishmaniasis

Increased reporting of CL case numbers in recent years suggests an increasing burden of disease within the region, a pattern in keeping with the forced displacement of Syrian refugees, particularly into Lebanon [66] and Turkey [67] in 2013. Broadly, increases in case reporting among local Turkish and Lebanese populations were not identified in this review despite their inclusion in many studies by design, possibly indicating minimal effect of mass refugee movement on local population VBD burden. This could also reflect challenges Syrian refugees face integrating into local societies, even when residing outside of camps [68].

Surges of CL cases in Syria occurred later (2014–2015), coinciding with increased violence and destruction of infrastructure in the hyper-endemic cities of Aleppo and Idlib in northwest Syria, as well as the highest number of attacks on medical facilities [69]. Decreasing CL case numbers following this period possibly to relate interventions including preventative measures from government and international health organizations, such as vector control initiatives and distribution of insecticide-treated nets [70]. While case numbers in some studies almost doubled between 2010 and 2018, the trend seen pre-conflict may indicate that the disease epidemiology was already evolving, and was only exacerbated by the conflict [25].

While CL has been a major public health risk and endemic in Syria for over 60 years, cases were previously largely restricted to rural areas of Damascus and Aleppo governorates. However, recent data demonstrates increasing spread outside of these areas and to non-endemic areas and countries [62], likely due to forced displacement of millions of Syrians both as IDPs and refugees. It is notable that the most affected governorates in neighbouring countries are proximal to the Syrian border and host the largest number of Syrian refugees, namely southeast Turkey and Beka'a province of Lebanon.

Increased vector breeding is also a factor. It has also been shown that cracks in buildings due to bombings, and build-up of domestic waste create optimal sand-fly breeding conditions [25,63]. Unfortunately, despite trends in reporting location, mass population displacement means accurate mapping of disease origin and true leishmaniasis burden is not possible. This, alongside the incubation period of leishmaniasis of up to eight months, mean no studies were able to accurately identify location and period of infection, confounding geographical interpretation [67].

## Healthcare access

Though this systematic review focused on clinical-epidemiological studies, some studies provided information about healthcare access including models of care and barriers to access. It is possible that cases, particularly among refugees, are affected by underreporting from poor health-seeking behaviour, barriers to access [14,25] or misdiagnosis [24,61], a deteriorating health system and compromised surveillance systems across the region. Conversely, overreporting due to population migration leading to duplicate reporting of patients when seeking continued care in a new location is also possible [25]. Exact case numbers and disease trends within Syria in particular, are therefore difficult to determine [42].

Outside of Syria, refugees face continued barriers to accessing healthcare. Refugees often settle in low-resource areas far from health services [19,70]. Furthermore, in Lebanon, residency policy makes obtaining legal status difficult [47]. Reports in 2018 showed 74% of Syrians in Lebanon lacked legal residency status, which is likely to severely restrict their access to healthcare. Despite this, the training of healthcare professionals, the creation of VBD-specific treatment centres

and public education is likely to have contributed to the control of CL in Lebanon and has been credited with positively impacting the incidence of other CDs which increase during conflict [14].

### *Leishmania* species

Species identification has several clinical benefits: determining the diagnosis, route of transmission (anthroponotic versus zoonotic) treatment duration and dosage, and prognosis; all these factors that differ between species [40]. *L.tropica* was the main causative species of CL identified across settings. Reports of classic VL pathogens (*L.donovani, L.infantum*) causing CL in Syria, mainly Idlib province, require further attention for vector control [24]. The identification of *L.aethiopica* species, never before seen in Turkey, shows further vector investigation is needed [24]. The identification of a new *Leishmania* species in Turkey is of concern, as this demonstrates possible introduction of isolates from Syria [42]. It is known that reservoir populations differ between *Leishmania* species, with *L.tropica* typically anthroponotic and *L.major* and *L. infantum* zoonotic species [40] Increasing numbers of reported cases caused by zoonotic species identified in this review suggest changes in reservoir populations in these regions, and show the need for animal studies to develop our understanding of CL epidemiology.

### Strengths and limitations

Limitations of this systematic review include the limited available academic literature on VBDs in general with most focused on leishmaniasis, particularly CL. However, the review provides a summary of available literature on leishmaniasis in Syria and the main Syrian refugee hosting countries providing some indications of the burden in these countries, though this may not correlate to real world epidemiology given that reporting, particularly in conflict, is likely to be poor. Our exclusion of grey literature and non-English language academic publications may mean that we have missed some relevant academic literature on VBDs however, in Syria, most academic literature is published in English or bilingually; the conflict has also negatively impacted Arabic-only Syrian journals which had existed before the conflict. Given the nature of the publications, a strength of this literature is its summation of findings on this topic together with gaps in the literature.

## Conclusions

This systematic review highlights the impact of conflict on leishmaniasis as well as a dearth of data in the academic literature on other VBDs potentially affected by the Syrian conflict. Emerging themes of importance were identified relating to healthcare access for refugees and models of surveillance or healthcare provided for IDPs and refugees. The lack of robust academic exploration of VBDs in Syria is stark after almost fifteen years of conflict [71–73]. Since the toppling of the Syrian regime in December 2024, there is an opportunity for better collaboration and exploration of existing, emerging and re-emerging VBDs (beyond leishmaniasis alone) given evolving factors which include climate change, population movements and breakdown in vector control measures due to funding interruptions [74,75]. For leishmaniasis, there remains important research gaps through a One Health approach which relate to the vectors, hosts, environment and other factors which can impact its surveillance, spread or control [76–78].

Further comparative analysis of incidence pre- and post-conflict onset will also allow for a broader description of the patterns of VBD within this region.

### Key findings

Leishmaniasis is the main vector-borne infection reported in academic literature among Syrians in Syria and the main surrounding refugee-hosting countries since the conflict

Studies report that the epidemiological burden of leishmania in Syria and surrounding refugee hosting countries has increased during the conflict.

Evidence gaps relate to non-leishmaniasis vector-borne infections, e.g., Crimean Congo Haemorrhagic Fever and insufficient information on effective interventions.

Barriers to healthcare access and breakdown in surveillance during the conflict are likely a key factor to the lack of reporting of vector-borne infections, something which must improve in the post-conflict period.

A One Health approach which explores environment, vectors, hosts and humans and the presence of vector-borne diseases is important in the post-conflict period, particularly due to changes related to climate, water and population movements.

## Supporting information

**S1 Appendix. Table of search strategy term across 4 chosen databases.**
(DOCX)

**S2 Appendix. Quality assessment of included papers.**
(DOCX)

**S3 Appendix. Details of included manuscripts.**
(XLSX)

**S1 File. PRISMA checklist [28].**
(DOCX)

## Author contributions

**Conceptualization:** Aula Abbara.

**Data curation:** Jack Carew, Remi Simpson, Timesh D. Pillay, Angel Desai, Richard Toalster, Pylin Parkes.

**Formal analysis:** Jack Carew, Remi Simpson, Timesh D. Pillay.

**Investigation:** Jack Carew.

**Methodology:** Remi Simpson, Timesh D. Pillay, Angel Desai, Richard Toalster, Pylin Parkes, Aula Abbara.

**Project administration:** Aula Abbara.

**Validation:** Jack Carew, Aula Abbara.

**Visualization:** Remi Simpson, Timesh D. Pillay.

**Writing – original draft:** Jack Carew, Remi Simpson, Timesh D. Pillay, Angel Desai, Richard Toalster, Pylin Parkes, Aula Abbara.

**Writing – review & editing:** Jack Carew, Richard Toalster, Pylin Parkes, Aula Abbara.

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
