## [Decision Letter · Decision Letter 0]

14 May 2025

PNTD-D-25-00006

Vector-borne diseases and the Syrian Conflict: A Systematic Review

Dear Dr. Abbara,

Thank you for submitting your manuscript to PLOS Neglected Tropical Diseases. After careful consideration, we feel that it has merit but does not fully meet PLOS Neglected Tropical Diseases's publication criteria as it currently stands. Therefore, we invite you to submit a revised version of the manuscript that addresses the points raised during the review process.

Please submit your revised manuscript within 60 days Jul 13 2025 11:59PM. If you will need more time than this to complete your revisions, please reply to this message or contact the journal office at plosntds@plos.org. Please include the following items when submitting your revised manuscript:

We look forward to receiving your revised manuscript.

Kind regards,

Brice Rotureau, PhD

Academic Editor

Susan Madison-Antenucci

Section Editor

Shaden Kamhawi

co-Editor-in-Chief

Paul Brindley

co-Editor-in-Chief

**Additional Editor Comments :**

Please, consider all reviewers' comments and modify the title so that it better reflects the content of this review.

**Journal Requirements:**

At this stage, the following Authors/Authors require contributions: John William Carew, Remi Simpson, Timesh D Pillay, Angel Desai, Richard Toalster, Pylin Parkes, and Aula Abbara. Please ensure that the full contributions of each author are acknowledged in the "Add/Edit/Remove Authors" section of our submission form.

5) We notice that your supplementary Tables, and information are included in the manuscript file. Please remove them and upload them with the file type 'Supporting Information'. Please ensure that each Supporting Information file has a legend listed in the manuscript after the references list.

Potential Copyright Issues:

i) Figures 1, and 3. Please (a) provide a direct link to the base layer of the map (i.e., the country or region border shape) and ensure this is also included in the figure legend; and (b) provide a link to the terms of use / license information for the base layer image or shapefile. We cannot publish proprietary or copyrighted maps (e.g. Google Maps, Mapquest) and the terms of use for your map base layer must be compatible with our CC BY 4.0 license.

ii) Appendix 2 contains a logo or branding. We are not permitted to publish this under our CC-BY 4.0 license, even with permission. We ask that you please remove or replace it.

7) We note that your Data Availability Statement is currently as follows: "All data is available in the manuscript". Please confirm at this time whether or not your submission contains all raw data required to replicate the results of your study. Authors must share the “minimal data set” for their submission. PLOS defines the minimal data set to consist of the data required to replicate all study findings reported in the article, as well as related metadata and methods (https://journals.plos.org/plosone/s/data-availability#loc-minimal-data-set-definition).

8) Please provide a completed 'Competing Interests' statement, including any COIs declared by your co-authors. If you have no competing interests to declare, please state "The authors have declared that no competing interests exist". Otherwise please declare all competing interests beginning with the statement "I have read the journal's policy and the authors of this manuscript have the following competing interests:"

9) Please upload the PRISMA flowchart  as Figure 1.

10) As required by our policy on Data Availability, please ensure your manuscript or supplementary information includes the following:

**Comments to the Authors:**

**Please note that one of the reviews is uploaded as an attachment.**

**Reviewers' Comments:**

Reviewer's Responses to Questions

PLOS authors have the option to publish the peer review history of their article (what does this mean?). If published, this will include your full peer review and any attached files.

Reviewer #1: No

Reviewer #2: No

Reviewer #3: No

**Key Review Criteria Required for Acceptance?**

**Methods**

-Are the objectives of the study clearly articulated with a clear testable hypothesis stated?

-Is the study design appropriate to address the stated objectives?

-Is the population clearly described and appropriate for the hypothesis being tested?

-Is the sample size sufficient to ensure adequate power to address the hypothesis being tested?

-Were correct statistical analysis used to support conclusions?

-Are there concerns about ethical or regulatory requirements being met?

Reviewer #2: The objectives are clearly stated in the abstract and introduction, focusing on exploring the epidemiological burden and geographical distribution of VBDs in Syria and neighboring countries following the conflict. However, the study is a systematic review, and thus does not have a hypothesis in the same way as an original research study. This should be acknowledged.

The systematic review design is appropriate for addressing the objectives. PRISMA guidelines were followed, which enhances the study's rigor.

The population (Syrian refugees and host countries) is clearly described and relevant to the hypothesis.

As this is a systematic review, "sample size" refers to the number of studies included. The authors included 33 studies, which appears to be a reasonable number for a systematic review. The review's strength lies in synthesizing the available data, not in statistical power calculations.

Systematic reviews primarily involve narrative synthesis of data. The authors extracted data and summarized it narratively, which is appropriate. They also assessed the quality of included studies using the EPHPP tool, which is a recognized method.

Ethical approval is not applicable for a systematic review of published studies.

Reviewer #3: I believe the closer co-operation with local experts and refugee camps will bring more needed results, as are presented in the current study. Inclusion criteria should be more broad, authors are missing important VBD such as e.g. Crimean-Congo hemorrhagic fever. We similar and more detailed work was published in 2021 https://www.ijidonline.com/article/S1201-9712(20)32169-X/pdf

**Results**

-Does the analysis presented match the analysis plan?

-Are the results clearly and completely presented?

-Are the figures (Tables, Images) of sufficient quality for clarity?

Reviewer #2: Yes, the results section clearly presents the findings of the literature review, including study characteristics, VBD-related factors, causes for increased burden, and barriers to healthcare access, consistent with the stated methods.

Results are clearly presented, but Tables 2 and 3 could be simplified for better readability.

Figures 1–3 and tables are informative but would benefit from higher resolution and clearer labels.

Reviewer #3: Analysis match the plan and results are clearly presented.

**Conclusions**

-Are the conclusions supported by the data presented?

-Are the limitations of analysis clearly described?

-Do the authors discuss how these data can be helpful to advance our understanding of the topic under study?

-Is public health relevance addressed?

Reviewer #2: Conclusions are supported by the data, emphasizing leishmaniasis as the primary VBD and highlighting gaps in research.

The exclusion of grey literature and non-English studies is noted, but the authors should discuss how this might bias findings.

The exclusion of grey literature and non-English studies is noted, but the authors should discuss how this might bias findings.

Yes, the public health relevance is evident throughout the manuscript. The review highlights the importance of understanding and addressing VBDs in conflict settings, with clear implications for healthcare planning and response.

Reviewer #3: Conclusions are supported by the presented data and public health relevance is addressed. However, the review is only partial, and the major limitation is the exclusion of non-English language journals and the lack of cooperation with local experts.

**Summary and General Comments**

Reviewer #3: The paper titler ,, Vector-borne diseases and the Syrian Conflict: A Systematic Review “ was focused on Syria and neighbouring, refugee-hosting countries, but we know a lot of refugees are hosted by EU member states such as Germany. To be accurate, I suggest the change of the title to ,, Vector-borne diseases and the Syrian Conflict: A Systematic Review focused on Syria and neighbouring, refugee-hosting countries”. Thanks to the inclusion criteria, the authors analysed 33 studies describing mainly leishmaniasis. The manuscript is well written with minor formal mistakes. However, I disagree with the authors' statement about the limitations of their study. Authors excluded non-English language academic publications and according to them,, this is unlikely to have a significant influence on the key information identified as most key, academic literature related to this topic is published in English.". A lot of important data has been published only in local journals written in the native language of practitioners from a non-academic environment. These are people working directly with refugees. However, as non-scientifically oriented health workers, they usually tend to publish their data only for the local experts in the local journals. Journals are usually published in the native language and are not listed in databases. We can see this trend even in EU countries. I believe this is the main limitation of the current study, describing not more than the already available data without direct communication with refugee camps. The main advantage of the current manuscript for me is not the need to read 33 studies, but only one, which is fine and fits the scope of the journal. However, I do not see any further scientific impact, and I will appreciate the authors admitting it, as a minimum, in the Strengths and limitations section. Moreover, similar work was published in 2021, https://www.ijidonline.com/article/S1201-9712(20)32169-X/pdf.

**Figure resubmission:**
---

## [Decision Letter · Decision Letter 1]

4 Nov 2025

Dear Dr. Abbara,

We are pleased to inform you that your manuscript 'Vector-borne diseases and the Syrian Conflict: A Systematic Review of literature from Syria and neighbouring, refugee-hosting countries' has been provisionally accepted for publication in PLOS Neglected Tropical Diseases.

Best regards,

Brice Rotureau, PhD

Academic Editor

Susan Madison-Antenucci

Section Editor

Shaden Kamhawi

co-Editor-in-Chief

Paul Brindley

co-Editor-in-Chief

Reviewer's Responses to Questions

PLOS authors have the option to publish the peer review history of their article (what does this mean?). If published, this will include your full peer review and any attached files.

Reviewer #1: No

Reviewer #2: **Yes: **Punya Ram Sukupayo, PhD

**Summary and General Comments**

Reviewer #1: All commends have been correctly addressed

---

## [Editor Report · Acceptance letter]

Dear Dr. Abbara,

We are delighted to inform you that your manuscript, "Vector-borne diseases and the Syrian Conflict: A Systematic Review of literature from Syria and neighbouring, refugee-hosting countries," has been formally accepted for publication in PLOS Neglected Tropical Diseases.

Best regards,

Shaden Kamhawi

co-Editor-in-Chief

Paul Brindley

co-Editor-in-Chief
